# Active hepatocellular carcinoma is an independent risk factor of direct-acting antiviral treatment failure: A retrospective study with prospectively collected data

**Yi-Hao Yen, Chien-Hung Chen, Chao-Hung Hung, Jing-Houng Wang, Sheng-Nan Lu, Kwong-Ming Kee, Tsung-Hui Hu** [ID] *

Department of Internal Medicine, Division of Hepatogastroenterology, Kaohsiung Chang Gung Memorial Hospital and Chang Gung University College of Medicine, Kaohsiung, Taiwan

* dr.hu@msa.hinet.net

**Data Availability Statement:** All relevant data are within the paper and its Supporting Information files.

## Abstract

### Background & aims

Previous studies from western countries have reported that active hepatocellular carcinoma (HCC) was associated with direct-acting antiviral (DAA) treatment failure. We sought to examine this issue in an Asian cohort.

### Methods

A retrospective cohort study was conducted on hepatitis C virus (HCV)-infected patients with advanced fibrosis who were treated with DAAs at our hospital between January 2017 and June 2018.

### Results

We treated 1021 HCV-infected patients during this period. A total of 976 of those patients were enrolled in a per-protocol analysis, including 556 (57.2%) who had genotype 1b infections, and 314 (32.3%) who had genotype 2 infections. The mean age of all 976 patients was 65.5 years, and 44.5% were male. 781 of the patients had no HCC, 172 had inactive HCC, and 23 had active HCC. Non-sustained virologic response (SVR) was noted in 10 (1.3%) patients without HCC, 5 (2.9%) patients with inactive HCC, and 4 (13.0%) patients with active HCC. After adjustment for confounders, active HCC (versus inactive HCC and non-HCC) was associated with non-SVR (adjusted odds ratio [AOR] = 24.5 (95% confidence interval [CI] = 4.4–136.9), P<0.001). Next, we excluded the 23 patients with active HCC from the multivariate analysis. After adjustment for confounders, inactive HCC (versus non-HCC) was not associated with non-SVR (AOR = 3.1 (95% CI = 0.94–9.95), P = 0.06).

**Funding:** This study was supported by Grant CMRPG8H1331 from the Kaohsiung Chang Gung Memorial Hospital, Taiwan. Grant Recipient is Yi-Hao Yen. The funders had no role in study design, data collection and analysis, decision to publish, or preparation of the manuscript.

**Competing interests:** The authors have declared that no competing interests exist.

## Conclusion

Active HCC was associated with non-SVR, while inactive HCC was not. We thus suggest the deferral of DAA treatment until after the complete radiological response of HCCs to treatment.

## Introduction

The availability of direct-acting antivirals (DAAs) has led to an increase in the number of patients receiving hepatitis C virus (HCV) treatment, including patients with hepatocellular carcinoma (HCC) [1]. The primary reason for treating HCV in patients with known HCC is similar to that for treating it in patients without HCC: to ameliorate the liver necroinflammation and fibrosis progression that can ultimately lead to the clinical consequences of cirrhosis [2].

A recent systematic review and meta-analysis reported that sustained virologic response (SVR) rates were lower in HCC patients treated with DAAs than in non-HCC patients treated with DAAs, especially in those with active HCC. However, the heterogeneity was high. Furthermore, the studies reviewed in the meta-analysis were all from western countries [1].

The aim of the present study was to examine whether active HCC was associated with DAA treatment failure in an Asian cohort.

## Patients and methods

### Patients

We performed a retrospective study that enrolled all HCV-infected patients with advanced fibrosis who were treated with DAAs at Kaohsiung Chang Gung Memorial Hospital between January 2017 and June 2018. The National Health Insurance Administration (NHIA) of Taiwan has provided reimbursements for DAAs since January 2017 for HCV-infected patients with advanced fibrosis. Advanced fibrosis was defined as the presence of any one of the following: transient elastography (TE) with a liver stiffness measurement (LSM) $\geq$ 9.5Kpa [3], a Fibrosis-4 (FIB-4) score $\geq$ 3.25 [4], a liver biopsy showing advanced fibrosis (METAVIR fibrosis score $\geq$ 3) [5], ultrasound-identified liver cirrhosis with splenomegaly, or endoscopy showing gastroesophageal varices.

The treatment regimens used for enrolled patients included daclatasvir+asunaprevir (DCV +ASV) for 24 weeks in genotype 1b patients without resistance-associated variants (RAVs) [6]; ombitasvir/paritaprevir/ritonavir and dasabuvir (3D) combined with ribavirin (RBV) for 12 weeks in genotype 1a patients without cirrhosis; 3D combined with RBV for 24 weeks in genotype 1a patients with cirrhosis; 3D for 12 weeks without RBV in genotype 1b patients; elbasvir/ grazoprevir (GZR/EBR) for 12 weeks without RBV in treatment-naïve genotype 1a patients without RAVs; GZR/EBR with RBV for 12 weeks in treatment-experienced genotype 1a patients without RAVs; 16 weeks of GZR/EBR combined with RBV for genotype 1a patients with non-structure protein 5A (NS5A) RAVs; 12 weeks of GZR/EBR without RBV in treatment-naïve genotype 1b patients; 12 weeks of GZR/EBR with RBV in treatment-experienced genotype 1b patients; 12 weeks of GZR/EBR without RBV in treatment-naïve genotype 4 patients; 16 weeks of GZR/EBR with RBV in treatment-experienced genotype 4 patients; sofosbuvir/ledipasvir (SOF/LDV) for 12 weeks without RBV in genotype 1, 4, 5, and 6 patients; SOF/LDV for 12 weeks with RBV in treatment-experienced, liver decompensation (Child–Turcotte–Pugh (CTP)B or C), or post-liver transplant patients; and SOF and RBV for 12

weeks for genotype 2 patients. In the case of suboptimal response to that last regimen [7], patients had the option to self-pay for and add DCV to the regimen. For our analysis, we classified the various regimens as either adequate or inadequate based upon HCV therapy guidelines [7]. Accordingly, SOF and RBV for 12 weeks for genotype 2 patients was defined as an inadequate regimen [7].

The presence of HCC in a patient was confirmed by histological or image analysis based on the recommendations of current guidelines [8–11].

As per the request of the NHIA, the treating physicians were required to input pre-treatment data (that is, ultrasound and lab data within 6 months of DAA initiation), as well as lab data at week 4 of the treatment, at the end of the treatment, and at week 12 of the follow-up period, into the national registry system. They were also required to input the reason (that is, intolerance, death, or other) for any premature discontinuation of treatment for patients who had such discontinuation, as well as the reason (that is, death or other) for not providing SVR12 data for any patients lacking such data. Due to the high costs of DAAs, the treating physicians were penalized if they did not input these data into the national registry system. Furthermore, all the patients signed an informed consent form provided by the NHIA which told them that they were required to comply with NHIA regulations; otherwise, their reimbursements for the DAA treatment would be canceled. As a result, there was ultimately only one patient who was still alive and without SVR data at the week 12 follow-up in this cohort. This 66-year-old female patient received a resection for HCC during the DAA treatment and then refused a follow-up appointment due to fatigue when a nurse contacted by phone (Table 1, case number 43).

We excluded 32 patients with premature discontinuation of treatment due to intolerance, as well as ten patients who died during treatment or before follow-up week 12. We also excluded 1 patient who was alive at follow-up week 12 but refused to make a follow-up appointment, 1 patient with HCC combined with cholangiocarcinoma, and 1 patient treated with DCV+ASV who was mistakenly not given a pre-treatment RAV test but was found to have RAVs (L31V, P58S, and Y93H in the HCV NS5A region) at the time of virologic relapse. Among the 45 excluded patients, 21 were patients with HCC (Table 1). Finally, a total of 976 patients were enrolled in this study.

Determination of the presence of an active tumor was based on the recommendations of current guidelines [8–11]. All other data was collected at the time of the initiation of DAA treatment and included the tumor size, tumor number, Barcelona Clinic Liver Cancer (BCLC) stage [10], and the treatment modalities received for HCC.

All the procedures used in the study were in accordance with the ethical standards of the responsible committees on human experimentation (institutional and national) and with the Helsinki Declaration of 1975, as revised in 2008. This study was approved by the Institutional Review Board of Kaohsiung Chang Gung Memorial Hospital (IRB number: 201801814B0). The requirement for informed consent was waived by the IRB. The data were analyzed anonymously.

## Definition of SVR

Serum HCV RNA levels were determined by COBAS TaqMan HCV Test (TaqMan HCV; Roche Molecular Systems Inc., Branchburg, N.J., lower limit of detection: 15 IU/ml). SVR was defined as undetectable HCV RNA 12 weeks after the end of therapy [12]. The genotyping of HCV was performed by primer-specific real-time PCR with the cobas® HCV GT assay (Roche Molecular Systems, Pleasanton, CA, USA).

**Table 1. Patients who were excluded from per protocol analysis.**

| Premature discontinue due to intolerance | Patient number | Gender | Age, years | CTP | HCCs | eGFR (ml/min/1.73$^2$) | Regimen | Reason of premature discontinue treatment | Duration of treatment (weeks) | SVR status |
|---|---|---|---|---|---|---|---|---|---|---|
| | 1 | M | 57 | A6 | N | 106 | DCV/ASV | Fever, diarrhea. | 16 | SVR |
| | 2 | F | 82 | A5 | Y | 52 | DCV/ASV | Post TACE syndrome | 18 | SVR |
| | 3 | F | 60 | A5 | N | 69 | 3D | Legs edema, dypsnea | 1 | Non-SVR |
| | 4 | F | 84 | A5 | Y | 80 | 3D | Hyperbilirubinemia, bilirubin (direct/total): 2.1/3.8 mg/dl | 1 | Non-SVR |
| | 5 | F | 84 | A5 | N | 42 | SOF/RBV | Fatigue | 4 | Unknown |
| | 6 | F | 83 | A5 | Y | 67 | GZR/EBR | Nausea | 4 | Unknown |
| | 7 | M | 84 | A5 | Y | 50 | SOF/RBV | Nausea | 8 | SVR |
| | 8 | M | 82 | A6 | Y | 28 | SOF/LDV+RBV | Fatigue | 4 | Unknown |
| | 9 | M | 81 | A5 | N | 60 | SOF/RBV | Epigastralgia | 4 | Unknown |
| | 10 | F | 81 | A5 | Y | 50 | 3D | Nausea | 3 | Unknown |
| | 11 | F | 81 | A5 | N | 62 | 3D | Nausea | 2 | Non-SVR |
| | 12 | F | 80 | A5 | N | 54 | 3D | Nausea | 2 | Non-SVR |
| | 13 | F | 79 | A6 | N | 100 | GZR/EBR | Delirium | 6 | SVR |
| | 14 | F | 75 | A5 | Y | 82 | 3D | Hyperbilirubinemia, bilirubin (direct/total): 3.3/5.8 mg/dl | 1 | Non-SVR |
| | 15 | M | 75 | A5 | Y | 97 | GZR/EBR | Prostate cancer with bone metastasis, wish hospice care | 4 | Non-SVR |
| | 16 | F | 75 | A5 | N | 55 | 3D | Palpitation | 3 | Non-SVR |
| | 17 | F | 74 | A5 | N | 54 | SOF/RBV | Palpitation | 2 | Non-SVR |
| | 18 | F | 74 | A5 | N | 10 | 3D | Renal function downhill | 1 | Non-SVR |
| | 19 | F | 74 | A5 | N | ESRD | GZR/EBR | Dizziness, nausea | 4 | SVR |
| | 20 | M | 74 | A5 | Y | 59 | 3D | Post TACE syndrome | 10 | Unknown |
| | 21 | F | 70 | A5 | Y | 154 | SOF/RBV | Hypertension, poor control | 2 | Non-SVR |
| | 22 | F | 69 | A5 | N | ESRD | 3D | Delirium | 2 | Non-SVR |
| | 23 | F | 69 | A5 | Y | 90 | 3D | Legs edema | 11 | SVR |
| | 24 | M | 68 | A5 | N | 57 | SOF/RBV | Syncope | 8 | Non-SVR |
| | 25 | M | 68 | A5 | Y | 15 | GZR/EBR | AST/ALT: 540/264 (U/L) | 2 | Non-SVR |
| | 26 | F | 66 | A5 | N | 97 | 3D | Palpitation, depression, fatigue | 3 | Non-SVR |
| | 27 | F | 64 | A5 | N | 93 | 3D | Stroke | 4 | Unknown |
| | 28 | M | 64 | A5 | N | 85 | 3D | Legs edema | 10 | SVR |
| | 29 | F | 58 | A5 | N | ESRD | 3D | Nausea, vomiting | 4 | Non-SVR |
| | 30 | M | 63 | A5 | N | 78 | 3D | Liver decompensation (new onset of ascites) | 8 | Non-SVR |
| | 31 | M | 38 | A5 | N | 126 | GZR/EBR | Ulcer bleeding | 10 | Non-SVR |
| | 32 | M | 59 | A5 | N | 8 | 3D | Liver decompensation, (new onset of ascites, variceal bleeding) | 2 | Non-SVR |
| Died during treatment Or before follow up week 12 | Patient number | Gender | Age, years | CTP | HCCs | eGFR (ml/min/1.73$^2$) | Regimen | Cause of death | Duration of treatment (weeks) | SVR status |
| | 33 | F | 68 | A5 | Y | 56 | 3D | influenza B infection/respiratory failure | 10 | Unknown |
| | 34 | F | 91 | A5 | N | 65 | DCV/ASV | Seizure/aspiration pneumonia | 3 | Unknown |
| | 35 | M | 66 | A5 | Y | 76 | DCV/ASV | Acute myocardial infarction | 14 | Unknown |
| | 36 | F | 59 | A5 | N | 55 | SOF/RBV | Ovary cancer with peritoneal carcinomatosis | 6 | Unknown |

*(Continued)*

**Table 1.** (Continued)

| Premature discontinue due to intolerance | Patient number | Gender | Age, years | CTP | HCCs | eGFR (ml/min/1.73²) | Regimen | Reason of premature discontinue treatment | Duration of treatment (weeks) | SVR status |
|---|---|---|---|---|---|---|---|---|---|---|
| | 37 | F | 78 | A5 | Y | 58 | 3D | Nausea, poor intake, pre-renal azotemia | 1 | Unknown |
| | 38 | M | 63 | C11 | N | 164 | SOF/LDV +RBV | Necrotizing fascitis | 5 | Unknown |
| | 39 | F | 87 | A6 | Y | 73 | SOF/RBV | HCCs with extrahepatic spread | 12 | Unknown |
| | 40 | M | 66 | A5 | Y | 15 | SOF/RBV | Staphylococcus aureus sepsis, decompensated cirrhosis | 12 | Unknown |
| | 41 | F | 77 | A5 | Y | 77 | GZR/EBR | variceal bleeding | 12 | Unknown |
| | 42 | F | 79 | B7 | Y | 41 | SOF/LDV/ RBV | Decompensated cirrhosis, pneumonia | 12 | Unknown |
| Miscellaneous | Patient number | Gender | Age, years | CTP | HCCs | eGFR (ml/min/1.73²) | Regimen | Reasons of exclusion | Treatment duration (weeks) | SVR status |
| | 43 | F | 66 | A5 | Y | 70 | GZR/EBR | Refuse follow due to fatigue after resection for HCC | 12 | Unknown |
| | 44 | M | 54 | A5 | Y | 76 | SOF/LDV/ RBV | HCC combined cholangiocarcinoma | 12 | SVR |
| | 45 | F | 62 | A5 | N | 110 | DCV/ASV | Did not check pre-treatment RAVs, virologic relapse with RAVs (L31V, P58S, Y93H in NS5A). | 24 | Relapse |

SVR, sustained virologic response; SOF, sofosbuvir; SOF/LDV, sofosbuvir plus ledipasvir; 3D, ritonavir-boosted paritaprevir, plus ombitasvir and dasabuvir; GZR/EBR, grazoprevir plus elbasvir; DCV, daclatasvir; ASV, Asunaprevir; RBV, ribavirin; HCC, hepatocellular carcinoma; eGFR, estimated Glomerular filtration rate; TACE, Transcatheter arterial chemoembolization; AST, aspartate aminotransferase; ALT, alanine aminotransferase; RAVs, resistant associated variants; NS5A, non-structure protein 5A;ESRD, end stage renal disease

## Statistical analysis

The baseline characteristics of the patients were summarized as mean (± standard deviation), median (interquartile range), or frequency (percentage). The distributions of the baseline characteristics according to the HCC and SVR status were estimated using the chi-squared or Fisher's exact test for categorical variables, and estimated using the independent two-sample t-test for continuous variables. Covariates in the multivariable model were chosen *a priori* for clinical importance. The potential confounders included age, gender, platelet count, prior history of interferon-based treatment, CTP class, and DAA regimen. Each *p*-value was two-sided and was considered statistically significant if the *p*-value less than 0.05. All analyses were performed using Stata version 14.0. (StataCorp. 2015. Stata Statistical Software: Release 14. College Station, TX: StataCorp LP.).

## Results

The baseline characteristics of and a comparison between the patients in the active HCC, inactive HCC and non-HCC groups in this cohort are shown in Table 2. There were 976 patients in the cohort, 781 of the patients had no HCC, 172 had inactive HCC, and 23 had active HCC. In this cohort with advanced fibrosis, only 28 (2.9%) patients had decompensated cirrhosis (defined by CTP class B or C). Genotype 1b and 2 patients accounted for 870 (89.1%) of the patients in the entire cohort, while 61 patients were genotype 1a and 40 were genotype 6. Meanwhile, none of the patients in the cohort were genotype 3 patients because

**Table 2. Baseline characteristics of HCV patients who underwent treatment with DAA stratified by HCC status.**

| Characteristics | Entire cohort, N = 976 | Non-HCC, N = 781 | Inactive HCC, N = 172 | Active HCC, N = 23 | P |
|---|---|---|---|---|---|
| Age (years) | 65.5 ± 10.1 | 64.5 ± 10.3 | 70.2 ± 7.9 | 65.9 ± 8.4 | <0.001 |
| Male | 435 (44.6%) | 341 (43.7%) | 80 (46.5%) | 14 (60.9%) | 0.224 |
| BMI (kg/m2) | 25 ± 4.0 | 25 ± 4.0 | 24.7 ± 4.0 | 25.5 ± 3.4 | 0.521 |
| Treatment regimen, n (%) | | | | | - |
| Daclatasvir+Asunaprevir, n (%) | 93 (9.5%) | 75 (9.6%) | 15 (8.7%) | 3 (13.0%) | |
| Harvoni, n (%) | 120 (12.3%) | 94 (12.0%) | 26 (15.1%) | 0 (0%) | |
| Harvoni+Rib, n (%) | 47 (4.8%) | 37 (4.7%) | 8 (4.7%) | 2 (8.7%) | |
| Sofosbuvir+Rib, n (%) | 266 (27.3%) | 217 (27.8%) | 45 (26.2%) | 4 (17.4%) | |
| Sofosbuvir+Rib+Daclatasvir, n (%) | 49 (5.0%) | 31 (4.0%) | 14 (8.1%) | 4 (17.4%) | |
| Viekirax+Dasabuvir, n (%) | 252 (25.8%) | 209 (26.8%) | 37 (21.5%) | 6 (26.1%) | |
| Viekirax+Dasabuvir+Rib, n (%) | 32 (3.3%) | 29 (3.7%) | 3 (1.7%) | 0 (0%) | |
| Zepatier, n(%) | 117 (12.0%) | 89 (11.4%) | 24 (14.0%) | 4 (17.4%) | |
| Creatinine (mg/dL) | 1.1 ± 1.4 | 1.1 ± 1.4 | 1.2 ± 1.4 | 1.3 ± 1.8 | 0.300 |
| AFP (ng/ml)* | 5.9 (3.4–12.5) | 5.6 (3.2–10.9) | 7.6 (4–15.1) | 92.2 (10–297.1) | <0.001 |
| Albumin (mg/dL) | 4.2 ± 0.4 | 4.2 ± 0.4 | 4.0 ± 0.4 | 3.7 ± 0.4 | <0.001 |
| AST (IU/L)* | 61 (43–97) | 60 (42–94) | 65 (43.5–95) | 116 (60–178) | <0.001 |
| ALT (IU/L)* | 69 (42–116) | 68 (41–116) | 67 (43–110.5) | 104 (65–145) | 0.250 |
| Total bilirubin (mg/dL) | 1.0 ± 0.6 | 1.0 ± 0.6 | 1.0 ± 0.5 | 1.2 ± 0.7 | 0.145 |
| Platelet (109/L)* | 131 (98–172) | 137 (102–177) | 115.5 (85–145) | 88 (60–99) | <0.001 |
| INR | 1.1 ± 0.3 | 1.1 ± 0.3 | 1.1 ± 0.1 | 1.1 ± 0.1 | 0.820 |
| HCV genotype | | | | | 0.275 |
| 1b | 556 (57.2%) | 441 (56.5%) | 100 (58.1%) | 15 (65.2%) | |
| 2 | 314 (32.3%) | 247 (31.6%) | 59 (34.3%) | 8 (34.8%) | |
| Others | 102 (10.5%) | 89 (11.4%) | 13 (7.6%) | 0 (0%) | |
| HCV RNA (log IU/ml) | 13.4 ± 2.1 | 13.5 ± 2.1 | 13.2 ± 2.1 | 13.4 ± 1.9 | 0.276 |
| Interferon experienced, n(%) | 274 (28.1%) | 224 (28.7%) | 46 (26.7%) | 4 (17.4%) | 0.500 |
| Final SVR, code = 1, n(%) | 957 (98.1%) | 770 (98.6%) | 167 (97.1%) | 20 (87.0%) | 0.003 |
| Ascites, n(%) | 11 (1.1%) | 9 (1.2%) | 0 (0%) | 2 (8.7%) | 0.016 |
| Decompensation, n(%) | 28 (2.9%) | 22 (2.8%) | 4 (2.3%) | 2 (8.7%) | 0.222 |
| HBsAg positive, n(%) | 57 (5.8%) | 56 (7.2%) | 1 (0.6%) | 0 (0%) | <0.001 |
| LT, n(%) | 9 (0.9%) | 8 (1.0%) | 1 (0.6%) | 0 (0%) | 1.000 |
| HCV-HIV coinfection, n(%) | 3 (0.3%) | 3 (0.4%) | 0 (0%) | 0 (0%) | 1.000 |
| APRI | 1.9 ± 1.8 | 1.8 ± 1.6 | 2.2 ± 2.0 | 4.8 ± 3.9 | <0.001 |
| FIB-4 | 5.0 ± 4.4 | 4.5 ± 3.7 | 6.3 ± 4.9 | 11.7 ± 11.2 | <0.001 |

P-value is estimated using chi-squared, Fisher's exact or one-way ANOVA test.

Data are presented as mean, standard deviation or number (%).

*AFP, AST, ALT and Platelet are presented as median (interquartile range)

SVR, sustained virologic response; BMI, body mass index; AFP, alpha-fetoprotein

SOF, sofosbuvir; SOF/LDV, sofosbuvir plus ledipasvir; 3D, ritonavir-boosted paritaprevir, plus ombitasvir and dasabuvir; GZR/EBR, grazoprevir plus elbasvir; DCV, daclatasvir; ASV, Asunaprevir; RBV, ribavirin; AST, aspartate aminotransferase; ALT, alanine aminotransferase; INR, international normalized ratio; HCV, hepatitis C virus; HBsAg, hepatitis B surface antigen; LT, liver transplantation; HIV, human immunodeficiency virus; HCC, hepatocellular carcinoma; APRI, aspartate aminotransferase-to-platelet ratio index; FIB-4, fibrosis-4 index

reimbursements were not being provided for the regimen for genotype 3 during this period. Compared to the inactive HCC and non-HCC patients, the patients with active HCC had higher alpha-fetoprotein (AFP), aspartate aminotransferase-to-platelet ratio index (APRI), and

FIB-4 levels; and had lower albumin and platelet levels, a lower SVR rate and higher proportion of patients with ascites.

## HCC patient characteristics

There were 195 patients with HCC. The mean age of these patients was 69.7 years, their median AFP level was 8.3 ng/ml at HCV treatment initiation, and 25.6% of the patients were treatment-experienced. Genotype 1b was the predominant genotype among these patients (59%), while the tumor characteristics of the HCC group at diagnosis are shown in Table 3. The average tumor size at HCC diagnosis was 2.6 ± 1.6 cm, and the majority of patients with a tumor present were at BCLC stage 0 or A (83.1%). Only 8 patients (4.1%) underwent liver transplantation (LT) with a pre-transplantation diagnosis of HCC, while recurrent HCC was not noted in any of those patients post-LT. Hepatic resection was performed in 57 (29.2%) patients, 143 (73.3%) patients received radiofrequency ablation (RFA), and 81 (41.5%) patients received transcatheter arterial chemoembolization (TACE).

## Characteristics of patients with active HCCs

The characteristics of the patients with active HCC are shown in Table 4. Twenty-three patients had active HCC at the initiation of DAA treatment. Among those patients, the tumor stage at the initiation of DAA treatment was BCLC stage B in 4 patients, BCLC stage C in 2 patients, and BCLC stage 0 or A in the remaining 17 patients. Only 3 of the patients did not achieve SVR, and 2 of those patients, both of whom had BCLC stage 0 (Table 4, case numbers 5 and 7), were treated with SOF/DCV/RBV. The third patient, who had BCLC stage C, was treated with SOF+RBV. He received concurrent sorafenib and DAA treatment (Table 4, case number 18).

**Table 3. Tumor characteristics of patients with HCC who underwent treatment with DAA.**

| HCC tumor characteristics | HCC group (n = 195) |
|---|---|
| Size (cm) | 2.6 ± 1.6 |
| Number | |
| 1, N (%) | 141 (72.3%) |
| 2–3, N (%) | 39 (20.0%) |
| 4 or more, N (%) | 7 (3.6%) |
| Unknown | 8 (4.1%) |
| BCLC | |
| 0, N (%) | 52 (26.7%) |
| A, N(%) | 110 (56.4%) |
| B, N(%) | 23 (11.8%) |
| C, N(%) | 5 (2.6%) |
| Unknown | 5 (2.6%) |
| Treatment received | |
| Resection, N (%) | 57 (29.2%) |
| Liver transplant, N (%) | 7 (3.6%) |
| RFA, N (%) | 143 (73.3%) |
| TACE, N (%) | 81 (41.5%) |
| Others, N (%) | 6 (3.1%) |

HCC, hepatocellular carcinoma; BCLC, Barcelona Clinic Liver Clinic; RFA, radiofrequency ablation; TACE, transcatheter arterial chemoembolization.

Patients may have received multiple therapies

**Table 4. Clinical characteristics of patients with active HCCs.**

| Patient number | Sex | Age, years | CTP | AFP (ng/ml) | Platelet count ($10^9$/L) | SVR | Regimen | Genotype | Interferon experienced | Tumor number | Tumor size (cm) | BCLC |
|---|---|---|---|---|---|---|---|---|---|---|---|---|
| 1 | M | 65 | A6 | 1098 | 72 | SVR | SOF/DCV/RBV | 2 | N | 1 | 2.2 | A |
| 2 | M | 68 | A6 | 341 | 143 | SVR | GZR/EBR | 1b | N | >10 | 2.3 | B |
| 3 | M | 52 | A6 | 8.6 | 149 | SVR | 3D | 1b | N | 3 | 1.9 | A |
| 4 | M | 70 | B9 | 209 | 91 | SVR | SOF/LDV/RBV | 1b | N | 1 | 1.5 | 0 |
| 5 | F | 72 | A5 | 7.8 | 131 | Non-SVR | SOF/DCV/RBV | 2 | N | 1 | 1.1 | 0 |
| 6 | M | 66 | A5 | 6.4 | 90 | SVR | SOF/RBV | 2 | N | 1 | 1.3 | 0 |
| 7 | F | 71 | A6 | 150 | 40 | Non-SVR | SOF/DCV/RBV | 2 | N | 1 | 1.6 | 0 |
| 8 | F | 58 | A5 | 92 | 86 | SVR | GZR/EBR | 1b | Y | 1 | 2 | 0 |
| 9 | F | 76 | A6 | 27 | 60 | SVR | GZR/EBR | 1b | N | 4 | 1.7 | B |
| 10 | M | 66 | A5 | 416 | 133 | SVR | 3D | 1b | N | 1 | 1.6 | 0 |
| 11 | F | 60 | B8 | 120 | 60 | SVR | SOF/LDV/RBV | 1b | N | 1 | 2.3 | C (Post RFA, no viable tumor in liver. A seeding tumor at abdominal wall). |
| 12 | F | 48 | A5 | 204 | 30 | SVR | 3D | 1b | N | 1 | 1.6 | 0 |
| 13 | F | 66 | A5 | 10 | 91 | SVR | SOF/RBV | 2 | N | 2 | 1 | A |
| 14 | M | 70 | A5 | 463 | 88 | SVR | GZR/EBR | 1b | N | 1 | 2 | A |
| 15 | F | 58 | A5 | 31 | 88 | SVR | 3D | 1b | Y | 2 | 1.8 | A |
| 16 | M | 74 | A5 | 77 | 92 | SVR | DCV/ASV | 1b | N | 2 | 1.4 | A |
| 17 | M | 67 | A5 | 122 | 61 | SVR | DCV/ASV | 1b | N | 5 | 1.1 | B |
| 18 | M | 61 | A6 | 10 | 55 | Non-SVR | SOF/RBV | 2 | N | 1 | 2.3 | C (Left portal vein tumor thrombus) |
| 19 | F | 79 | A5 | 10 | 112 | SVR | SOF/DCV/RBV | 2 | N | 2 | 1.5 | A |
| 20 | M | 59 | A5 | 44 | 58 | SVR | SOF/RBV | 2 | N | 2 | 1.5 | A |
| 21 | M | 72 | A5 | 1044 | 63 | SVR | DCV/ASV | 1b | Y | >5 | 3 | B |
| 22 | M | 75 | A5 | 15 | 76 | SVR | 3D | 1b | Y | 1 | 1.8 | 0 |
| 23 | M | 52 | A5 | 297 | 99 | SVR | 3D | 1b | N | 1 | 1.5 | 0 |

HCCs, hepatocellular carcinomas; CTP, Child–Turcotte–Pugh; LSM, liver stiffness measurement by transient elastography; SOF, sofosbuvir; SOF/LDV, sofosbuvir plus ledipasvir; 3D, ritonavir-boosted paritaprevir, plus ombitasvir and dasabuvir; GZR/EBR, grazoprevir plus elbasvir; DCV, daclatasvir; ASV, Asunaprevir; RBV, ribavirin; BCLC, Barcelona Clinic Liver Cancer; RFA, radiofrequency ablation

### Non-SVR rate by regimen

The HCV treatment regimens used for all the patients are shown in Table 2. SOF/RBV for 12 weeks was the most common regimen (27%), followed by 3D for 12 weeks (25.8%), LDV/SOF (12.3%) for 12 weeks, and GZR/EBR (12.0%) for 12 weeks. Thirty-two (3.3%) patients were treated with 3D+RBV (the treatment duration was 12 weeks in 12 patients, 24 weeks in 20 patients); all of those patients were genotype 1a and all achieved SVR. No patients received more than 12 weeks of GZR/EBR treatment.

The patients with HCC were compared to those without HCC regarding specific DAA regimens (Fig 1). For patients treated with DCV/ASV, non-SVR was noted in 1 HCC patient (5.6%), while all of the non-HCC patients achieved SVR. For patients treated with LDV/SOF, all of the HCC patients achieved SVR, while non-SVR was noted in one (1.1%) of the non-HCC patients. For patients treated with LDV/SOF+RBV and 3D+RBV, all of the patients

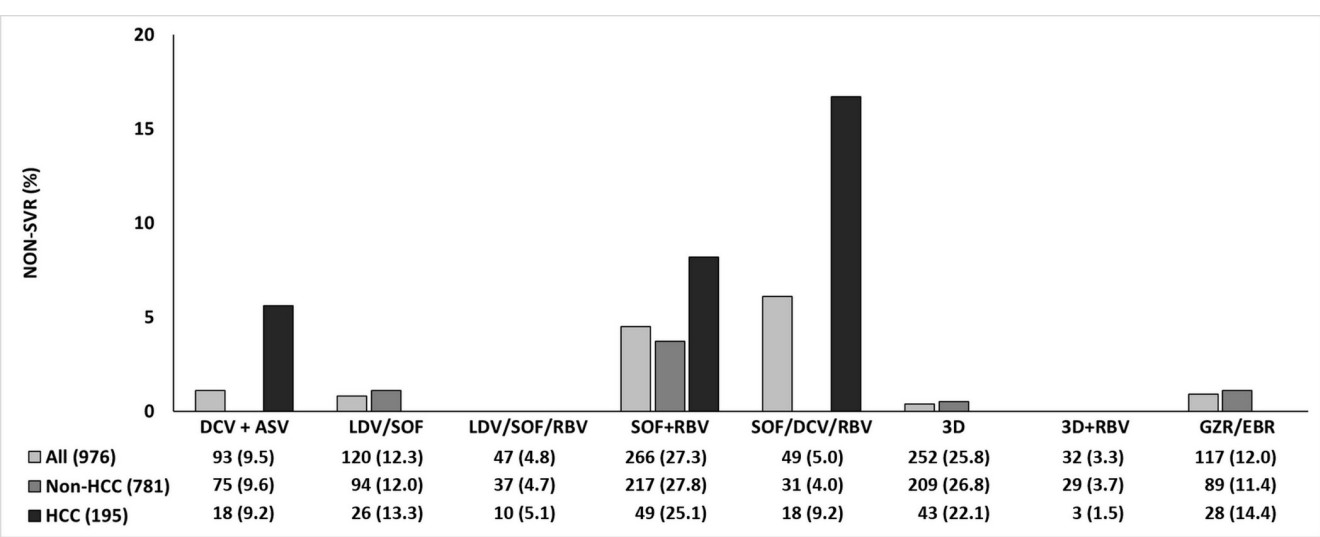

**Fig 1. Non-SVR rate of DAA therapy by treatment regimen received.** SOF, sofosbuvir; SOF/LDV, sofosbuvir plus ledipasvir; 3D, ritonavir-boosted paritaprevir, plus ombitasvir and dasabuvir; GZR/EBR, grazoprevir plus elbasvir; DCV, daclatasvir; ASV, Asunaprevir;

(including all of the HCC and non-HCC patients) achieved SVR. For patients treated with SOF+DCV+ RBV, non-SVR was noted in 3 (16.7%) HCC patients, while all of the non-HCC patients achieved SVR. For patients treated with SOF/RBV, non-SVR was noted in 4 HCC patients (8.2%), while non-SVR was noted in 8 (3.7%) of the non-HCC patients. For patients treated with 3D, all of the HCC patients achieved SVR, while non-SVR was noted in 1 (0.5%) of the non-HCC patients. For patients treated with GZR/EBR, all of the HCC patients achieved SVR, while non-SVR was noted in 1 (1.1%) of the non-HCC patients.

The comparison of SVR rates between patients with and without HCC stratified by genotype and treatment regimen were shown in S1 Table. The SVR rates were not significantly different between patients with and without HCC stratified by genotype and treatment regimen except in genotype 2 patients treated with SOF+DCV+RBV for 12 weeks.

### Clinical characteristic of patients with non-SVR

The clinical characteristic of the patients with non-SVR are shown in Table 5. Twelve patients were treated with SOF and RBV for 12 weeks. Among these patients, 5 patients had an LSM >13ka (the cutoff value of METAVIR F4) [13–15], and 2 patients had active HCC. Seven patients were treated with a regimen other than SOF and RBV. Among these patients, 4 patients had an LSM >13 kPa, and 2 patients had active HCC.

### Univariate predictors of non-SVR

The univariate predictors of non-SVR are shown in Table 6. The proportion of patients with genotype 2, proportion of patients with a history of HCC, proportion of patients with active HCC, and proportion of patients who were treated with the inadequate regimen (SOF+RBV) were higher in the non-SVR group; AST levels were higher in the non-SVR group.

### Multivariable predictors of non-SVR

The multivariable predictors of non-SVR are shown in Table 7, the data for which were provided through Model A of our per-protocol (PP) analysis. There were 781 patients without

**Table 5. Characteristics of patients with non-SVR.**

| Patient number | Sex | Age, years | CTP | HCCs | LSM (kPa) | Platelet count ($10^9$/L) | Splenomegaly | Regimen | Genotype | Interferon experienced |
|---|---|---|---|---|---|---|---|---|---|---|
| 1 | F | 51 | A5 | N | 11.5 | 186 | N | GZR/EBR | 1b | N |
| 2 | M | 60 | A5 | Y, inactive, post liver transplant | NA | 270 | N | SOF/RBV | 2 | N |
| 3 | F | 72 | A5 | N | NA | 100 | Y | SOF/RBV | 2 | Y |
| 4 | M | 61 | A5 | Y, active | 21.5 | 55 | Y | SOF/RBV | 2 | N |
| 5 | M | 52 | A5 | N | 21.5 | 246 | s/p splenectomy | SOF/RBV | 2 | N |
| 6 | F | 72 | A6 | Y, active | NA | 40 | Y | SOF/RBV/DCV | 2 | N |
| 7 | M | 61 | A5 | Y, inactive | 27 | 105 | N | SOF/RBV/DCV | 2 | Y |
| 8 | F | 73 | A5 | Y, active | 22 | 131 | N | SOF/RBV/DCV | 2 | N |
| 9 | F | 80 | A5 | N | 45 | 121 | N | SOF/RBV | 2 | N |
| 10 | M | 82 | A5 | Y, active | 40 | 130 | Y | SOF/RBV | 2 | N |
| 11 | M | 59 | A5 | N | 27 | 127 | Y | SOF/LDV | 6 | N |
| 12 | F | 64 | A5 | N | 27 | 62 | Y | SOF/RBV | 2 | N |
| 13 | F | 67 | A5 | N | 12 | 132 | N | SOF/RBV | 2 | Y |
| 14 | F | 65 | A5 | N | 7.8 | 152 | N | SOF/RBV | 2 | N |
| 15 | M | 59 | A5 | Y, inactive | 15.5 | 158 | N | SOF/RBV | 2 | Y |
| 16 | M | 66 | A5 | N | 21.3 | 170 | N | 3D | 1b | Y |
| 17 | M | 58 | A5 | Y, inactive | NA | 118 | Y | SOF/RBV | 2 | N |
| 18 | M | 63 | A5 | N | 7.8 | 82 | Y | SOF/RBV | 2 | Y |
| 19 | F | 55 | A6 | Y, inactive | NA | 227 | N | DCV/ASV | 1b | N |

HCCs, hepatocellular carcinomas; CTP, Child–Turcotte–Pugh; LSM, liver stiffness measurement by transient elastography; SOF, sofosbuvir; SOF/LDV, sofosbuvir plus ledipasvir; 3D, ritonavir-boosted paritaprevir, plus ombitasvir and dasabuvir; GZR/EBR, grazoprevir plus elbasvir; DCV, daclatasvir; ASV, Asunaprevir; RBV, ribavirin; NA, not available; Y, yes; N, No

HCC, 172 patients with inactive HCC, and 23 patients with active HCC enrolled in the multivariate analysis. Non-SVR was noted in 5 (2.9%) patients with inactive HCC, 4 (13.0%) patients with active HCC, and 10 (1.3%) patients without HCC. After adjustment for confounders, active HCC (versus inactive HCC and non-HCC) was associated with non-SVR (adjusted odds ratio [AOR]: 24.5 (95% confidence interval [CI]: 4.4–136.9), P<0.001).

In Model B of our PP analysis, we excluded the 23 patients with active HCC; there were thus 781 patients without HCC and 172 patients with a history of inactive HCC enrolled into this multivariate analysis. After adjustment for confounders, a history of inactive HCC was not associated with non-SVR (AOR: 3.1(95% CI = 0.94–9.95), P = 0.062).

Model C consisted of an intention to treat (ITT) analysis. We excluded 1 patient with HCC combined with cholangiocarcinoma and 1 patient with virologic relapse due to malpractice (Table 1, case numbers 44 and 45). We then enrolled 43 patients who were initially excluded from the PP analysis (Table 1, case numbers 1–43) into this analysis. Overall, there were 215 patients with HCC and 804 patients without HCC included in the analysis. Non-SVR was noted in 24 (11.2%) of the patients with HCC and 28 (3.5%) of the patients without HCC. After adjustment for confounders, HCC (AOR: 2.8(95% CI: 1.5–5.2), P = 0.001) was associated with non-SVR.

**Table 6. Univariate predictors of non-SVR.**

| Characteristics | SVR, N = 957 | Non-SVR, N = 19 | P |
|---|---|---|---|
| Age (years) | 65.5 ± 10.1 | 64.4 ± 8.5 | 0.48 |
| Male | 425 (44.4%) | 10 (52.6%) | 0.61 |
| BMI (kg/m$^2$) | 25.0 ± 4.0 | 25.6 ± 5.1 | 0.55 |
| Treatment regimen, n (%) | | | 0.01 |
| DCV/ASV, n (%) | 92 (9.6%) | 1 (5.3%) | |
| SOF/LDV, n (%) | 119 (12.4%) | 1 (5.3%) | |
| SOF/LDV+RBV, n (%) | 47 (4.9%) | 0 (0.0%) | |
| SOF+RBV, n (%) | 254 (26.5%) | 12 (63.2%) | |
| SOF+DCV+RBV, n (%) | 46 (4.8%) | 3 (15.8%) | |
| 3D, n (%) | 251 (26.2%) | 1 (5.3%) | |
| 3D+RBV, n (%) | 32 (3.3%) | 0 (0.0%) | |
| GZR/EBR+RBV, n (%) | - | - | |
| GZR/EBR, n(%) | 114 (11.9%) | 3 (15.8%) | |
| Creatinine (mg/dl) | 1.1 ± 1.4 | 0.8 ± 0.2 | 0.41 |
| AFP (ng/ml) | 5.9 (3.4–12.5) | 6.5 (4.2–24.2) | 0.92 |
| Albumin (mg/dl) | 4.2 ± 0.4 | 4.1 ± 0.5 | 0.34 |
| AST (IU/L) | 61 (43–95) | 83 (44–162) | 0.03 |
| ALT (IU/L) | 68 (42–114) | 101 (52–145) | 0.16 |
| Total bilirubin (mg/dl) | 1.0 ± 0.6 | 1.0 ± 0.5 | 0.72 |
| Platelet (10$^9$/L) | 131 (98–172) | 130 (100–170) | 0.97 |
| INR | 1.1 ± 0.3 | 1.1 ± 0.1 | 0.77 |
| HCV genotype | | | <0.001 |
| 1b | 553 (57.8%) | 3 (15.8%) | |
| 2 | 299 (31.2%) | 15 (78.9%) | |
| Others | 101 (10.6%) | 1 (5.3%) | |
| HCV RNA (log IU/ml) | 13.4 ± 2.1 | 13.9 ± 2.6 | 0.35 |
| Interferon experienced, n(%) | 268 (28.0%) | 6 (31.6%) | 0.73 |
| Ascites, n(%) | 11 (1.1%) | 0 (0.0%) | 0.81 |
| Decompensation, n(%) | 28 (2.9%) | 0 (0.0%) | 0.45 |
| HBsAg positive, n(%) | 55 (5.7%) | 2 (10.5%) | 0.31 |
| LT, n(%) | 14 (1.5%) | 1 (5.3%) | 0.16 |
| HCV-HIV coinfection, n(%) | 3 (0.3%) | 0 (0.0%) | 0.94 |
| HCC, n(%) | 187 (19.5%) | 8 (42.1%) | 0.02 |
| Active HCC, n(%) | 20 (2.1%) | 3 (15.8%) | 0.009 |

SVR, sustained virologic response; BMI, body mass index; AFP, alpha-fetoprotein

SOF, sofosbuvir; SOF/LDV, sofosbuvir plus ledipasvir; 3D, ritonavir-boosted paritaprevir, plus ombitasvir and dasabuvir; GZR/EBR, grazoprevir plus elbasvir; DCV, daclatasvir; ASV, Asunaprevir; RBV, ribavirin; AST, aspartate aminotransferase; ALT, alanine aminotransferase; INR, international normalized ratio; HCV, hepatitis C virus; HBsAg, hepatitis B surface antigen; LT, liver transplantation; HIV, human immunodeficiency virus; HCC, hepatocellular carcinoma.

## Discussion

Several factors are reportedly associated with DAA treatment failure, including cirrhosis, inadequate drug regimens, and adherence [16–21]. Regarding cirrhosis, Prenner et al. conducted a retrospective study on cirrhotic patients who were treated with DAAs. In that study, cirrhosis was defined by one of the following: liver biopsy, TE >12.5 kPa, acoustic radiation force impulse (ARFI) >2.0 m/s, magnetic resonance elastography >5 kPa, or FibroSURE[TM] testing [22]. Among these non-invasive tests, only TE is available in our hospital. However, TE can be

**Table 7. Multivariable predictors of non-SVR.**

| Covariate | Model A | | | Model B | | | Model C | | |
|---|---|---|---|---|---|---|---|---|---|
| | OR | 95% CI | *P* | OR | 95% CI | *P* | OR | 95% CI | *P* |
| Age (per year) | 0.98 | 0.93–1.02 | 0.35 | 0.96 | 0.91–1.01 | 0.12 | 1.03 | 1.00–1.07 | 0.05 |
| Gender, male vs. female | 1.24 | 0.47–3.25 | 0.67 | 1.47 | 0.51–4.28 | 0.48 | 0.79 | 0.43–1.44 | 0.44 |
| Decompensated cirrhosis, yes vs. no | - | | | - | | | 1.88 | 0.40–8.84 | 0.43 |
| Platelet, <100 vs. ≥100 ($10^9$/L) | 0.38 | 0.09–1.55 | 0.18 | 0.41 | 0.09–1.90 | 0.26 | 0.68 | 0.34–1.36 | 0.28 |
| HCC*, yes vs. no | 24.47 | 4.37–136.93 | <0.001 | 3.07 | 0.94–9.95 | 0.06 | 2.82 | 1.53–5.20 | 0.001 |
| Treatment regimen, SOF+RBV vs. others | 6.79 | 2.44–18.84 | <0.001 | 8.5 | 2.76–26.21 | <0.001 | 1.66 | 0.91–3.04 | 0.100 |
| Interferon experienced, yes vs.no | 1.96 | 0.70–5.52 | 0.20 | 2.17 | 0.74–6.32 | 0.16 | 1.05 | 0.54–2.04 | 0.88 |

Model A: per protocol analysis. HCC*: active HCC versus inactive HCC and non-HCC. All patients with decompensated cirrhosis achieved SVR

Model B: per protocol analysis. HCC*: inactive HCC versus non-HCC. All patients with decompensated cirrhosis achieved SVR

Model C: intention to treat analysis, HCC*: HCC versus non-HCC

HCC, hepatocellular carcinoma; SVR, sustained virologic response; SOF, sofosbuvir; RBV, ribavirin.

inaccurate in HCC patients with tumors located at the right lobe of the liver or who have undergone right hepatectomy. Few patients underwent liver biopsy in our cohort, and histology results were available mainly for those who underwent resection for HCC. Therefore, we did not include cirrhosis as a covariate in the multivariate analysis. Instead, we used substages and substage indicators of cirrhosis such as platelet count <100 ($10^9$/L)(surrogate marker of clinical significant portal hypertension)[23] and decompensated cirrhosis as covariates in the multivariate analysis. Regarding inadequate regimens, SOF/RBV for 12 weeks is an inadequate regimen for genotype 2 patients with cirrhosis [7]. Therefore, SOF/RBV versus other regimens was included as a covariate in the multivariate analysis. Regarding adherence, we used a PP analysis in this study.

Active HCC was associated with non-SVR by the PP analysis. The possible mechanisms include the possibilities that HCC may lead to distortion of the liver architecture and decreased DAA delivery and that HCC may function as a reservoir for HCV replication [24, 25].

A history of inactive HCC was not associated with non-SVR according to the PP analysis conducted in our study. In contrast, a previous study reported that a history of inactive HCC was associated with DAA treatment failure. However, the authors of that study did not mention whether their result was based on an ITT or PP analysis [26].

Furthermore, we performed an ITT analysis. We enrolled 43 patients who were initially excluded from the PP analysis (Table 1, case numbers 1–43) into this ITT analysis. Among those 43 patients, 20 were patients with HCC, and non-SVR was noted in 17 of these patients with HCC. Of the remaining 195 patients with HCC who completed DAA treatment, non-SVR was noted in only 8 patients. Therefore, HCC was associated with DAA treatment failure mainly due to the intolerance of DAA treatments.

Regarding specific DAA regimens in HCC patients, the non-SVR rate was highest in the patients treated with SOF+DCV+RBV for 12 weeks. Three (16.7%) patients were non-SVR after being treated with this regimen (Table 5, case numbers 6–8), two of the non-SVR patients had active HCC, and all of the non-SVR patients had clinically significant portal hypertension defined by either platelet count < 100 ($10^9$/L) and splenomegaly or LSM> 20kPa [23, 27]. Real-world data from Taiwan have shown high SVR rates with this regimen in genotype 2 patients with advanced fibrosis (98.5% and 100%, respectively) [28, 29]. Therefore, the higher non-SVR rate with this regimen in our study was due to advanced cirrhosis and active HCC. The non-SVR rate was the second highest in patients treated with SOF/RBV for 12 weeks.

Four (8.2%) patients were non-SVR after being treated with this regimen. 100% SVR was noted in HCC patients treated with LDV/SOF, 3D, and GZR/EBR, although the numbers of such patients were limited.

Prenner et al. conducted a retrospective cohort study that enrolled cirrhotic patients treated with DAA in a LT center [22]. In their multivariable analysis, active HCC at the time of DAA initiation was associated with non-SVR. That result was compatible with our study. However, there were also some differences between Prenner's study and our study. Firstly, more advanced liver disease was noted in Prenner's study; all of the patients were cirrhotic and 26% of the patients had decompensated cirrhosis. In contrast, while all of the patients in our study had advanced fibrosis, only 2.9% had decompensated cirrhosis. Secondly, a higher proportion of patients were treated with inadequate regimens such as SOF/simeprevir for 12 weeks (46%) in Prenner's study. In contrast, an inadequate regimen (SOF/RBV for 12 weeks) was used to treat only 27.3% of the patients in our study. This difference could explain the higher non-SVR rate in Prenner's (14.7%) study compared with our study (1.9%).

In another study, Beste, et al. examined SVR rates among veterans with and without HCC. In that study, the rate of SVR was 91.9% in non-HCC patients, 74.5% in HCC patients, and 93.4% in patients with a pre-LT diagnosis of HCC who underwent LT. This data was abstracted from a corporate data warehouse, with each diagnosis of HCC being obtained using the International Classification of Diseases (ICD) codes. Therefore, the number of patients with active HCC could not be evaluated in that study. Meanwhile, the patients in the HCC group who were treated with DAAs after LT had similar failure rates to those without HCC. Based on these findings, Beste, et al. recommended that the deferral of DAA treatment until the post-LT setting may be considered among HCC patients listed for LT [30]. However, due to the extreme shortage of deceased donors in Taiwan, the deferral of DAA treatment until the post-LT setting in not feasible in Taiwan.

In terms of clinical application, the findings of our study include several key points: first, the patients with HCC were older and had more advanced liver disease, and the association of HCC with non-SVR was mainly due to the HCC patients being more intolerant of DAA treatment. Second, active HCC was associated with non-SVR in the PP analysis, while inactive HCC was not. We thus recommend that DAA treatment be commenced after a complete radiological response to HCC treatment has been achieved. Although liver decompensation is the major driver of death in HCV-related HCC patients [31], no evidence supports the conclusion that patients with active HCC gain a survival benefit after DAA treatment. Current guidelines also recommend DAA treatment in those who have undergone curative treatment for HCC [12].

The strength of this study is that it was a retrospective study with prospectively collected data. Due to the high cost of DAAs, the patients and physicians in Taiwan were informed that they must comply with the regulations of the NHIA. Therefore, only one patient who was alive at follow-up week 12 with an unknown SVR outcome was noted in this cohort, and there was no missing data for the cohort enrolled in the PP analysis. Secondly, we comprehensively examined the possible mechanisms for HCC with DAA treatment failure. According to our ITT analysis, HCC was associated with non-SVR due to the HCC patients being more intolerant of DAA treatment. According to our PP analysis, active HCC was associated with non-SVR, while inactive HCC was not.

Our study had several limitations. Firstly, around 40% of the patients in our cohort received first generation all-oral DAA regimens (DCV/ASV, SOF/RBV, SOF/DCV/RBV), which are no longer recommended by current guidelines [12]. Future studies with the currently recommended DAA therapies will thus be needed to confirm the findings of the present study. Secondly, the most important cofounder associated with non-SVR is the presence of cirrhosis [12]. However, TE is not feasible in patients with HCC. Therefore, we recommend that ARFI,

which has a high diagnostic accuracy, be used to evaluate cirrhosis in patients with or without HCC in future studies [32]. Thirdly, the reimbursements from the NHIA in Taiwan allow us to aggressively treat HCV-infected patients, including patients with active HCC or a limited life expectancy. The findings of this study thus may not be generalizable to other countries that only treat patients as recommended by the guideline [12].

In conclusion, in this study, the patients with HCC were older and had more advanced liver disease, which led them to be relatively intolerant of DAA treatment and caused a lower SVR rate. Furthermore, active HCC was associated with non-SVR, while inactive HCC was not, so we suggest the deferral of DAA treatment until after complete radiological response to HCC treatment has been achieved.

## Supporting information

**S1 Data. raw data.**
(XLSX)

**S1 Table. The comparison of SVR rates between patients with and without HCC stratified by genotype and treatment regimen.**
(DOCX)

## Author Contributions

**Conceptualization:** Yi-Hao Yen.

**Data curation:** Yi-Hao Yen.

**Formal analysis:** Yi-Hao Yen.

**Funding acquisition:** Yi-Hao Yen.

**Investigation:** Yi-Hao Yen.

**Methodology:** Yi-Hao Yen.

**Project administration:** Yi-Hao Yen.

**Resources:** Yi-Hao Yen.

**Supervision:** Yi-Hao Yen, Chien-Hung Chen, Chao-Hung Hung, Jing-Houng Wang, Sheng-Nan Lu, Kwong-Ming Kee, Tsung-Hui Hu.

**Validation:** Yi-Hao Yen.

**Visualization:** Yi-Hao Yen.

**Writing – original draft:** Yi-Hao Yen.

**Writing – review & editing:** Yi-Hao Yen.

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
