## [Decision Letter · Decision Letter 0]

28 Jul 2019

PONE-D-19-19092

Active hepatocellular carcinoma is an independent risk factor of direct-acting antiviral treatment failure: a retrospective study with prospectively collected data

PLOS ONE

Dear Dr Hu,

Thank you for submitting your manuscript to PLOS ONE. After careful consideration, we feel that it has merit but does not fully meet PLOS ONE’s publication criteria as it currently stands. Therefore, we invite you to submit a revised version of the manuscript that addresses the points raised by two reviewers during the review process.

We would appreciate receiving your revised manuscript by Sep 11 2019 11:59PM. To enhance the reproducibility of your results, we recommend that if applicable you deposit your laboratory protocols in protocols.io, where a protocol can be assigned its own identifier (DOI) such that it can be cited independently in the future. For instructions see: http://journals.plos.org/plosone/s/submission-guidelines#loc-laboratory-protocols

We look forward to receiving your revised manuscript.

Kind regards,

Wenyu Lin, PhD

Academic Editor

PLOS ONE

Journal Requirements:

1. Thank you for including the following funding information within the acknowledgements section of your manuscript; "This study was supported by Grant CMRPG8H1331 from the Chang Gung Memorial

Hospital-Kaohsiung Medical Center, Taiwan. The funders had no role in study design,

data collection and interpretation, or the decision to submit the work for publication."

"The funders had no role in study design, data collection and analysis, decision to publish, or preparation of the manuscript"

Reviewers' comments:

Reviewer's Responses to Questions

**Comments to the Author**

1. Is the manuscript technically sound, and do the data support the conclusions?

Reviewer #1: Partly

Reviewer #2: Yes

2. Has the statistical analysis been performed appropriately and rigorously? 

Reviewer #1: I Don't Know

Reviewer #2: Yes

3. Have the authors made all data underlying the findings in their manuscript fully available?

Reviewer #1: Yes

Reviewer #2: Yes

4. Is the manuscript presented in an intelligible fashion and written in standard English?

Reviewer #1: Yes

Reviewer #2: Yes

5. Review Comments to the Author

Reviewer #1: The study aimed to investigate the association of active hepatocellular carcinoma (HCC) with DAA treatment failure. There are some problems in this study and below are my comments:

1. There are many active HCV related HCC patients (hundreds) in the hospital, but the authors only enroll 23 patients with active HCCs.DAA is not recommened for HCV with active HCC if life span was expected to be less than 6 months, how do the authors select which active HCC patients with HCV receive DAA. The selection bias may impact the result of the study.

2. The authors should compare the difference of baseline characters in non-HCC, inactive HCC and active HCC patients and revised in Table 2.

3. The study had as high as 41.5% of HCC patients received TACE before DAA Tx. TACE usually resulted in a low complete response rates in HCC patients. This in unexpected, how do the authors explain the high percentage of TACE here?

4. In table 6, the percentage of SVR vs non-SVR should be revised in several variates including tx regimen, genotype, HCC, active HCC etc.

5. In table 6, the authors should also include cirrhosis status ( not only decomepnsated) and fibrosis stage ( eg,using FIB-4) into univariate analysis and if significant, add into multivariate analyais.

6. Decompensated cirrhosis is believed to be associated with lower SVR in many studies, but why pts with decompensated cirrhosis have 100% SVR in this study.

7. In table 7. Some were analyzed in per protocol tx some in intention to treat, both per protocol and intention to treat should be analyzed in different group of patients.

8. In table 7, why genotype which is significant in univariate analysis is not included in multivariate analysis. Only 26.5% of patients received SOF/RBV, why the authors chose SOF/RBV vs non SOF/RBV in multivariate analysis in stead of SOF based vs non-SOF based regimen.

9. 172 had inactive HCC, and 23 had active HCC in this study. But in table 7, there were 215 patients with HCC and 804 patients without HCC included in the analysis. How do the authors explain the difference.

Reviewer #2: The author thoroughly analyzed the prediction power of active HCC on the success of anti-virus treatment. The data is well organized and presented in an logic way. There are no major concern about the paper, just a few suggestions that might be helpful

1 Any immunological reason that cause failure of DAAs? most of the HCC patients are very weak and through intense treatment, would the author supply more data about their immune system?

2 Would the comparison be carried between non-HCC and HCC with same serum type and same treatment? This might be more precise to define the prediction power of HCC, exclude confounding problem caused by serum type and treatment regime.

6. PLOS authors have the option to publish the peer review history of their article (what does this mean?). If published, this will include your full peer review and any attached files.

Reviewer #1: No

Reviewer #2: No

---

## [Author Response · Author response to Decision Letter 0]

6 Aug 2019

Reviewer #1: The study aimed to investigate the association of active hepatocellular carcinoma (HCC) with DAA treatment failure. There are some problems in this study and below are my comments:

1. There are many active HCV related HCC patients (hundreds) in the hospital, but the authors only enroll 23 patients with active HCCs. DAA is not recommened for HCV with active HCC if life span was expected to be less than 6 months, how do the authors select which active HCC patients with HCV receive DAA. The selection bias may impact the result of the study.

Response: Thank you so much for your comments. The National Health Insurance Administration (NHIA) of Taiwan has provided reimbursements for DAAs since January 2017 for HCV-infected patients with advanced fibrosis. Advanced fibrosis was defined as the presence of any one of the following: transient elastography (TE) with a liver stiffness measurement (LSM) ≥ 9.5Kpa [3], a Fibrosis-4 (FIB-4) score ≥ 3.25 [4], a liver biopsy showing advanced fibrosis (METAVIR fibrosis score ≥ 3) [5], ultrasound-identified liver cirrhosis with splenomegaly, or endoscopy showing gastroesophageal varices. Please see page 5, 1st paragraph. Patients with active HCC were not excluded from reimbursement if they had advanced fibrosis. 

2. The authors should compare the difference of baseline characters in non-HCC, inactive HCC and active HCC patients and revised in Table 2.

Response: Thank you so much for your comments. We have compare the difference of baseline characters in non-HCC, inactive HCC and active HCC patients and revised in Table 2. Compared to the inactive HCC and non-HCC patients, the patients with active HCC had higher alpha-fetoprotein (AFP), aspartate aminotransferase-to-platelet ratio index (APRI), and FIB-4 levels; and had lower albumin and platelet levels, a lower SVR rate and higher proportion of patients with ascites. Please see page 19, last 5 lines.

3. The study had as high as 41.5% of HCC patients received TACE before DAA Tx. TACE usually resulted in a low complete response rates in HCC patients. This in unexpected, how do the authors explain the high percentage of TACE here?

Response: Thank you so much for your comments. In our daily practice, patients may have received multiple therapies. We performed radiofrequency ablation for patients received TACE with incomplete radiological response. 

4. In table 6, the percentage of SVR vs non-SVR should be revised in several variates including tx regimen, genotype, HCC, active HCC etc.

Response: Thank you so much for your comments. We have provided the percentage of SVR vs non-SVR in several variates including tx regimen, genotype, HCC, active HCC in table 6.

5. In table 6, the authors should also include cirrhosis status ( not only decomepnsated) and fibrosis stage ( eg,using FIB-4) into univariate analysis and if significant, add into multivariate analyais.

Response: Thank you so much for your comments. Prenner et al. conducted a retrospective study on cirrhotic patients who were treated with DAAs. In that study, cirrhosis was defined by one of the following: liver biopsy, TE >12.5 kPa, acoustic radiation force impulse (ARFI) >2.0 m/s, magnetic resonance elastography >5 kPa, or FibroSURETM testing [22]. Among these non-invasive tests, only TE is available in our hospital. However, TE can be inaccurate in HCC patients with tumors located at the right lobe of the liver or who have undergone right hepatectomy. Few patients underwent liver biopsy in our cohort, and histology results were available mainly for those who underwent resection for HCC. Therefore, we did not include cirrhosis as a covariate in the multivariate analysis. Please see page 46, 1st paragraph.

In our previous studies enrolled more than 1700 treatment-naïve chronic hepatitis C patients who underwent liver biopsy prior to interferon therapy. Using liver biopsy as reference, the diagnostic accuracy of AST to Platelet Ratio Index (APRI) and FIB-4 to predict advanced fibrosis (≥F3) and cirrhosis (F4) are suboptimal (area under receiver operating characteristic curve (AUROC) around 0.70) [1], the diagnostic accuracy of 

ultrasound-identified cirrhosis is poor (AUROC=0.66) [2]. 

Further, the use of APRI and FIB-4 entails a risk of overestimating the fibrosis stage due to the impact of necroinflammatory activity on transaminases. In our previous study, we stratified the enrolled patients into the categories of AST≤37 IU/L (N=132), 37<AST≤74 IU/L, (N=501), 74<AST≤148 IU/L (N=737), and AST>148 IU/L (N=346). The upper limit for normal AST in our hospital is 37 IU/L. The cutoff values of FIB-4 to predict ≥F3 are 1.4, 2.2, 3.2, and 5.2 in the categories of AST≤37 IU/L, 37<AST≤74 IU/L, 74<AST≤148 IU/L, and AST>148 IU/L. Significant variations in the cutoff values of FIB-4 to predict ≥F3 were noted in patients stratified by AST level [1].

Reference: 

1. Yen YH, Kuo FY, Kee KM, Chang KC, Tsai MC, Hu TH, et al. APRI and FIB-4 in the evaluation of liver fibrosis in chronic hepatitis C patients stratified by AST level. PLoS One. 2018 Jun 28;13(6):e0199760.

2. Yen YH, Kuo FY, Chen CH, Hu TH, Lu SN, Wang JH, et al. Ultrasound is highly specific in diagnosing compensated cirrhosis in chronic hepatitis C patients in real world clinical practice. Medicine (Baltimore). 2019 Jul;98(27):e16270. 

6. Decompensated cirrhosis is believed to be associated with lower SVR in many studies, but why pts with decompensated cirrhosis have 100% SVR in this study.

Response: Thank you so much for your comments. The lower SVR rates in patients with decompensated cirrhosis as compared to patients with compensated cirrhosis in other studies were due to treatment discontinuations rather than virological failures [1]. In our study, two patients with decompensated cirrhosis received DAA treatment died during treatment or before follow up week 12, the SVR outcome of both patients were unknown. Please see table 1, case number 38 and 42. Twenty-eight patients with decompensated cirrhosis completed the treatment course and follow up and all patients achieved SVR. 

Reference: 1. European Association for the Study of the Liver. EASL Recommendations on Treatment of Hepatitis C 2018. J Hepatol. 2018;69:461-511

7. In table 7. Some were analyzed in per protocol tx some in intention to treat, both per protocol and intention to treat should be analyzed in different group of patients.

Response: Thank you so much for your comments.

Multivariable Predictors of Non-SVR were shown in Table 7. Covariates in the multivariable model were chosen a priori for clinical importance. The potential confounders included age, gender, platelet count, prior history of interferon-based treatment, CTP class, and DAA regimen. Please see page 18, 2nd paragraph, line 5-8. 

Several factors are reportedly associated with DAA treatment failure, including cirrhosis, inadequate drug regimens, and adherence [16-21]. Regarding cirrhosis, Prenner et al. conducted a retrospective study on cirrhotic patients who were treated with DAAs. In that study, cirrhosis was defined by one of the following: liver biopsy, TE >12.5 kPa, acoustic radiation force impulse (ARFI) >2.0 m/s, magnetic resonance elastography >5 kPa, or FibroSURETM testing [22]. Among these non-invasive tests, only TE is available in our hospital. However, TE can be inaccurate in HCC patients with tumors located at the right lobe of the liver or who have undergone right hepatectomy. Few patients underwent liver biopsy in our cohort, and histology results were available mainly for those who underwent resection for HCC. Therefore, we did not include cirrhosis as a covariate in the multivariate analysis. Instead, we used substages and substage indicators of cirrhosis such as platelet count <100 (109/L)［surrogate marker of clinical significant portal hypertension］[23] and decompensated cirrhosis as covariates in the multivariate analysis. Regarding inadequate regimens, SOF/RBV for 12 weeks is an inadequate regimen for genotype 2 patients with cirrhosis [7]. Therefore, SOF/RBV versus other regimens was included as a covariate in the multivariate analysis. Regarding adherence, we used a PP analysis in this study. Please see page 46. 

8. In table 7, why genotype which is significant in univariate analysis is not included in multivariate analysis. Only 26.5% of patients received SOF/RBV, why the authors chose SOF/RBV vs non SOF/RBV in multivariate analysis in stead of SOF based vs non-SOF based regimen.

Response: Thank you so much for your comments. Covariates in the multivariable model were chosen a priori for clinical importance. The potential confounders included age, gender, platelet count, prior history of interferon-based treatment, CTP class, and DAA regimen. Please see page 18, 2nd paragraph, line 5-8. Genotype is significant in univariate analysis of non-SVR is due to most of the non-SVR cases were genotype 2 patients who received inadequate regimen (i.e. SOF/RBV 12 weeks in cirrhotic patients). In contrast, SOF/DCV/RBV is not in inadequate regimen. Therefore, we did not chose SOF based vs non-SOF based regimen in multivariate analysis. 

9. 172 had inactive HCC, and 23 had active HCC in this study. But in table 7, there were 215 patients with HCC and 804 patients without HCC included in the analysis. How do the authors explain the difference.

Response: Thank you so much for your comments.

the data for which were provided through Model A of our per-protocol (PP) analysis. There were 781 patients without HCC, 172 patients with inactive HCC, and 23 patients with active HCC enrolled in the multivariate analysis. Please see page 42, first paragraph. 

Model C consisted of an I ntention to treat (ITT) analysis. We excluded 1 patient with HCC combined with cholangiocarcinoma and 1 patient with virologic relapse due to malpractice (Table 1, case numbers 44 and 45). We then enrolled 43 patients who were initially excluded from the PP analysis (Table 1, case numbers 1-43) into this analysis. Overall, there were 215 patients with HCC and 804 patients without HCC included in the analysis. Please see page 42 last paragraph and page 43 first paragraph.

Reviewer #2: The author thoroughly analyzed the prediction power of active HCC on the success of anti-virus treatment. The data is well organized and presented in an logic way. There are no major concern about the paper, just a few suggestions that might be helpful

1 Any immunological reason that cause failure of DAAs? most of the HCC patients are very weak and through intense treatment, would the author supply more data about their immune system?

Response: Thank you so much for your comments.

From genome-wide associated studies, single-nucleotide polymorphisms (SNPs) near the interleukin (IL) 28B locus have also shown association with treatment response with pegylated interferon and ribavirin therapy in patients with genotype 1 hepatitis C [1]. Slightly higher SVR rates have been seen in patients with the favorable IL28B genotype CC compared with those with TT in interferon-free DAA trials [2]; however, appropriately powered studies designed to assess this are still lacking. Our study is a retrospective study using chart review, we do not have data about their immune system.

References: 

1. Ge D, Fellay J, Thompson AJ, Simon JS, Shianna KV, Urban TJ, et al. Genetic variation in IL28B predicts hepatitis C treatment-induced viral clearance. Nature 2009;461:399–401.

2. Lawitz E, Sulkowski MS, Ghalib R, Rodriguez-Torres M, Younossi ZM, Corregidor A,et al. Simeprevir plus sofosbuvir, with or without ribavirin, to treat chronic infection with hepatitis C virus genotype 1 in nonresponders to pegylated interferon and ribavirin and treatment-naive patients: the COSMOS randomised study. Lancet 2014;384:1756–65.

2 Would the comparison be carried between non-HCC and HCC with same serum type and same treatment? This might be more precise to define the prediction power of HCC, exclude confounding problem caused by serum type and treatment regime.

 Response: Thank you so much for your comments. The comparison of SVR rates between patients with and without HCC stratified by genotype and treatment regimen were shown in supplementary table 1. The SVR rates were not significantly different between patients with and without HCC stratified by genotype and treatment regimen except in genotype 2 patients treated with SOF+DCV+RBV for 12 weeks.

---

## [Decision Letter · Decision Letter 1]

4 Sep 2019

[EXSCINDED]

Active hepatocellular carcinoma is an independent risk factor of direct-acting antiviral treatment failure: a retrospective study with prospectively collected data

PONE-D-19-19092R1

Dear Dr. Hu,

We are pleased to inform you that your manuscript has been judged scientifically suitable for publication and will be formally accepted for publication once it complies with all outstanding technical requirements.

With kind regards,

Wenyu Lin, PhD

Academic Editor

PLOS ONE

Additional Editor Comments (optional):

The authors have adequately addressed the comments raised by two reviewers. The manuscript is suitable to publish in Plos One.

Reviewers' comments:

Reviewer's Responses to Questions

**Comments to the Author**

1. If the authors have adequately addressed your comments raised in a previous round of review and you feel that this manuscript is now acceptable for publication, you may indicate that here to bypass the “Comments to the Author” section, enter your conflict of interest statement in the “Confidential to Editor” section, and submit your "Accept" recommendation.

Reviewer #2: All comments have been addressed

2. Is the manuscript technically sound, and do the data support the conclusions?

Reviewer #2: Yes

3. Has the statistical analysis been performed appropriately and rigorously? 

Reviewer #2: Yes

4. Have the authors made all data underlying the findings in their manuscript fully available?

Reviewer #2: Yes

5. Is the manuscript presented in an intelligible fashion and written in standard English?

Reviewer #2: Yes

6. Review Comments to the Author

Reviewer #2: the author well addressed all concerns rised by the reveiwers, data is solid and logically organized, the manuscripts meets publication criterias, should be suggested to publish.

7. PLOS authors have the option to publish the peer review history of their article (what does this mean?). If published, this will include your full peer review and any attached files.

Reviewer #2: No

---

## [Editor Report · Acceptance letter]

10 Sep 2019

PONE-D-19-19092R1 

Active hepatocellular carcinoma is an independent risk factor of direct-acting antiviral treatment failure: a retrospective study with prospectively collected data 

Dear Dr. Hu:

I am pleased to inform you that your manuscript has been deemed suitable for publication in PLOS ONE. Congratulations! Your manuscript is now with our production department. 

With kind regards,

on behalf of

Dr. Wenyu Lin 

Academic Editor

PLOS ONE